civil engineering/environmental engineering

lithology, broken waste rock, particle size distribution, strain, porosity, backfill mining

**Author for correspondence:**
Jixiong Zhang
e-mail: zjxiong@cumt.edu.cn

# An experimental study of the influence of lithology on compaction behaviour of broken waste rock in coal mine backfill

Meng Li[1], Jixiong Zhang[1], Zhongya Wu[1], Yu Liu[2] and Ailing Li[1]

[1]State Key Laboratory of Coal Resources and Safe Mining, School of Mines, China University of Mining and Technology, Xuzhou, Jiangsu 221116, People's Republic of China
[2]School of Mechanic Engineering, Jiangsu Normal University, Jiangsu, People's Republic of China

ML, 0000-0003-4789-2416; JZ, 0000-0003-1774-448X; YL, 0000-0003-1676-4446

The research aims to explore the influences of lithology on the compaction behaviours of broken waste rocks. For this purpose, a WAW-1000D servo test machine and a self-made bidirectional loading test system for granular materials were used to conduct axial and lateral compaction tests on four typical types of broken waste rocks: sandstone, mudstone, limestone and shale. On this basis, we analysed the relationships between lateral and axial stress with the strain in, and porosity of, the four types of broken waste rocks. In addition, the relationship of axial stress with lateral stress and lateral pressure coefficient, and the changes in the particle size distribution of broken waste rocks before and after compaction were discussed. The test results demonstrated that the samples of higher strength were found to have low lateral and axial strains as well as a lower porosity in axial and lateral loading tests, while samples of lower strength showed low lateral stress and lateral pressure coefficient under axial load. After being compacted, the samples of the four types of broken waste rocks were found to have a higher proportion of small particles, indicating some particle crushing. Moreover, the samples of lower strength were broken to a greater extent.

## 1. Introduction

The constant and high-intensity mining of coal resources has prompted the rapid development of China's economy, while at

the same time, it also leads to severe ecological problems such as surface subsidence, air pollution and water loss [1–5]. The production of coal inevitably results in the accumulation of huge amounts of waste rock at ground level, which not only occupies large areas, but also contaminates air and groundwater [6–10]. To solve a series of problems brought about by the discharge of waste rocks, scholars have proposed filling goafs with waste rock [11–13]. In this method, waste rocks are broken and then directly transported to the goaf; after applying a certain lateral pressure, the waste rocks for backfill reach a certain density in advance and are made to make full contact with the roof. After placing broken waste rock into the goaf, the rocks are gradually compacted and deformed under the action of overlying rocks. Therefore, the compaction behaviours of waste rocks play a decisive role in the effectiveness of goaf backfill [14–17].

Broken waste rocks are commonly used as backfill materials in solid backfill mining, so their compaction behaviours affect strata movement and surface subsidence in backfill mining. Scholars have investigated the compaction behaviours of broken waste rocks to good effect, for example, combining the testing machine with the self-made compaction device, Li *et al.* [18] studied the influences of particle size on the compaction behaviours of backfill materials. They concluded that broken waste rocks with large particle sizes constitute the skeleton structure of the backfill materials, while small particles fill the voids between larger particles, and a reasonable distribution of large and small particles is conducive to improving the stiffness of waste rock used for backfill. Pappas & Mark [19] investigated the compaction behaviours of rock caving in goafs and obtained the relationship between tangent and secant moduli and stress and described the stress–strain relationship using the Salamon and Terzaghi formulae. Zhang *et al.* [20,21] used a steel drum to test the crushing of particles of broken rocks with three particle sizes (15–20, 20–25 and 25–30 mm) in compaction under different loading pressures. By using a self-developed steel drum for compaction, Su *et al.* [22] performed compaction tests using an RMT-150B electro-hydraulic servo-controlled test system. In this way, they obtained the stress–strain relationship for broken rocks in a compaction test and analysed the influences of rock strength, block size and compaction stress on the compaction behaviours of the broken rocks. To study the fractal features of broken waste rocks in the compaction process, Zhang *et al.* [23] proposed a fractal model for the compaction and crushing of waste rocks and conducted compaction test under different compaction stresses and particle size distributions. By doing so, they verified that the theoretical model is reasonable. Li *et al.* [24] proposed a calculation method for the energy released during compaction of broken waste rocks and explored the influence of particle size on the dissipation of energy using a self-made compaction device and a SANS-CMT5305 test system. By using an MTS815.02 test system and a self-made test device, Ma *et al.* [25,26] tested the compaction behaviours of limestone particles and revealed the influence of particle size distribution on the changes in porosity during the compaction. Despite this work, to our knowledge, the influences of lithology on the compaction behaviours of broken waste rocks have yet been examined, especially when the effects of lateral pressure on the broken waste rock are considered.

Considering this, the preparation method of samples of broken waste rocks was introduced at first. Then, by testing the compaction behaviours of the broken waste rocks using the self-developed bidirectional loading test system for granular materials, the influence of lithology on the compaction behaviours of the broken waste rocks was studied. Meanwhile, the effects of changes in lateral strain, axial strain, porosity, lateral stress and lateral pressure coefficient of the samples under lateral and axial load were analysed, and the relationship between the lithology and compaction behaviours of broken waste rocks was established.

# 2. Sample preparation and test equipment

## 2.1. Sample preparation

The test materials were waste rocks collected from a coal mine, comprising four typical types of rocks: sandstone, mudstone, limestone and shale. Before the test, these four types of rocks were broken. At first, they were broken to 50 mm down and then broken again to 30 mm down, then graded to 5, 10, 15, 20, 25 and 30 mm, respectively, as shown in figure 1.

The test schemes for compaction of samples of different rock types are listed in table 1.

In the test schemes, the particle size distribution, lateral stress and number of cycles of lateral compaction were all kept unchanged to analyse the influences of lithology on the compaction behaviours of the broken waste rocks by taking different types of rocks, which was equivalent to

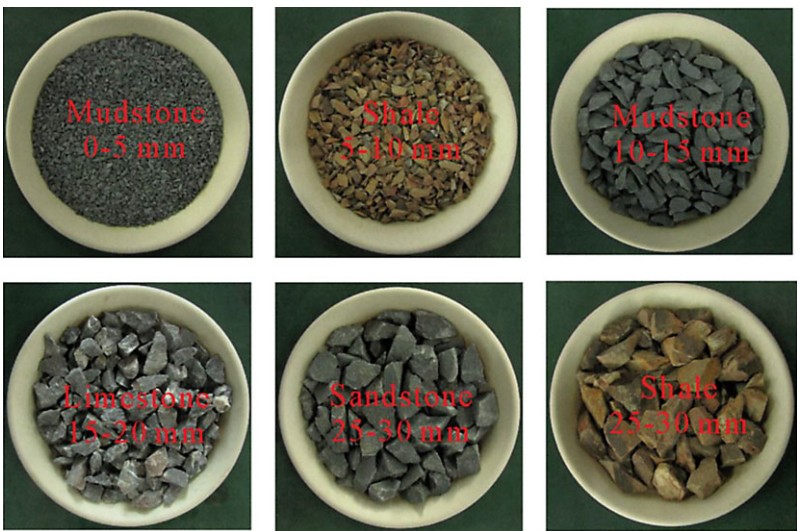

**Figure 1.** Samples of four types of broken waste rocks with different particle size.

**Table 1.** Test schemes for compaction of broken waste rocks.

| lithology | particle size distribution (mm) | lateral stress (MPa) | times of lateral compaction |
|---|---|---|---|
| sandstone | 0–30 | 2 | 5 |
| mudstone | 0–30 | 2 | 5 |
| limestone | 0–30 | 2 | 5 |
| shale | 0–30 | 2 | 5 |

indirectly adjusting the strength of samples. The particle size, lateral stress and number of cycles of lateral compaction were set to 0–30 mm, 2 MPa and 5, respectively. Meanwhile, to reduce the influence of test errors, three groups of tests were carried out for each scheme.

Considering the influences of particle size of broken waste rocks and to realize a compromise between the mass of broken waste rocks and the simplicity of test, each group of the samples for the four types of waste rock was prepared by uniformly mixing particles in the mass ratio of 1 : 1 by particle size.

## 2.2. Design of the test equipment

A bidirectional loading test system for granular materials was developed. The system was composed of four parts: an axial loading system, a loading box, a lateral loading system and a data monitoring and acquisition system, as displayed in figure 2.

### 2.2.1. Axial loading system

The axial loading system is the foundation of the bidirectional loading test system and provides constant and stable axial stress to granular materials. In the test, a WAW-1000D electro-hydraulic servo-controlled universal testing machine was used, which is capable of providing axial load to 1000 kN over a travel of 250 mm. In addition, being equipped with a large test bench, the machine can meet the requirements for tests of the compaction behaviours of granular materials.

### 2.2.2. Loading box

The loading box forms a loading chamber for granular materials and the location for fixing the oil cylinder for loading. It consists of a base, a ribbed slab, an arc-shaped support, a front plate, the plates on the left and right, an upper cover plate, a side-push plate and M18 bolts.

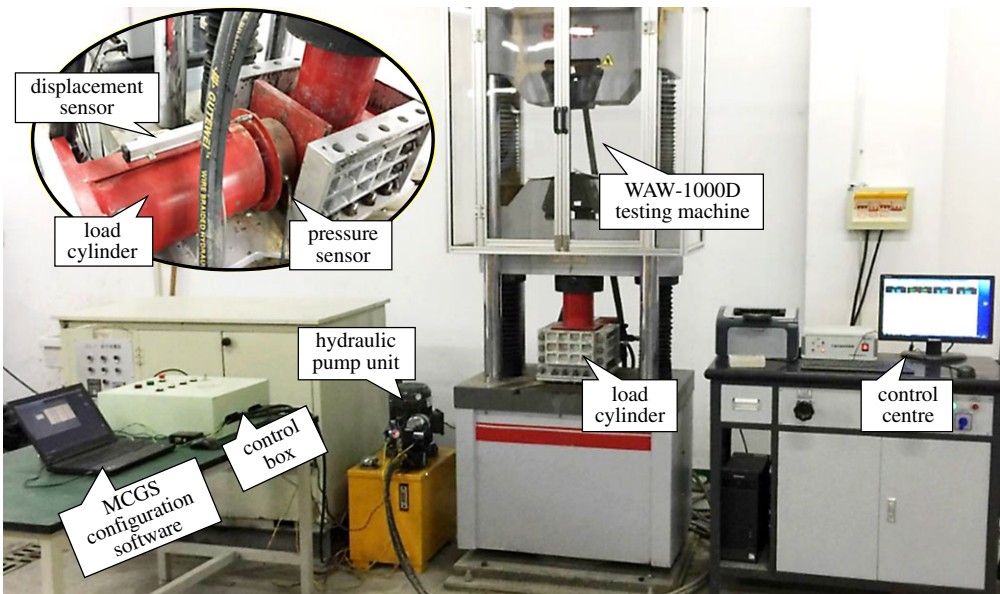

**Figure 2.** The bidirectional loading test system.

### 2.2.3. Lateral loading system

The lateral loading system provides lateral loading pressure on the samples of granular materials and includes a hydraulic pump unit, an oil cylinder for loading, a control box, a pressure gauge, a hydraulic pipe and an overflow valve.

### 2.2.4. Data monitoring and acquisition system

The data monitoring and acquisition system is able to monitor and acquire experimental data during loading. It is composed of a pull-rod displacement sensor, a spoke-structure pressure sensor, a pressure transmitter, FX2N-5a analogue input and output modules, an FX2N-32MR controller, MCGS configuration software and a laptop.

# 3. Test method and procedure

To study the influences of lithology on the compaction behaviours of broken waste rocks, compaction tests were carried out on the four typical types of waste rock (sandstone, mudstone, limestone and shale) using the developed bidirectional loading test system for granular materials. The influence of lithology on the compaction behaviours of broken waste rocks were obtained by following the test steps. The test method and procedure refer to the standard 'Method of compaction testing of solid backfilling materials' issued by China's National Energy Administration [27].

## 3.1. Preparing samples of broken waste rocks

The preparation scheme for the samples is summarized in §2.1.

## 3.2. Layering the prepared samples of broken waste rocks in the loading box

The loading box was assembled before placing the samples in and the displacement and pressure sensors were reset. Afterwards, the prepared samples were put in the loading box in three to six layers. After loading each layer, the surface of each layer was not very smooth; thus, the samples needed to be pre-compacted. Pre-compaction can make the surface of each layer smooth, and the experimental error can be greatly reduced. The total height of the samples placed in the loading box was 200 mm and the mass of the loaded samples in each group was recorded. After filling the box, bolts were used to fix the upper cover plate to the box. Then, the loading box was put on the test bench of the WAW-1000D electro-hydraulic servo-controlled testing machine.

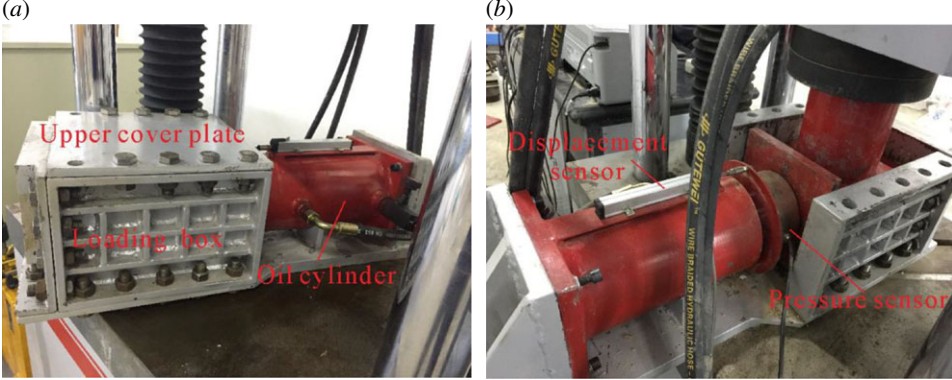

**Figure 3.** Lateral and axial loading of the samples. (*a*) Lateral loading and (*b*) axial loading.

## 3.3. Lateral loading

The motor of the hydraulic pump unit, the sensors and the PLC controller were powered on and the MCGS software was run to monitor the pressure and displacement. The required oil pressure, which was calculated corresponding to the lateral stress set according to the test scheme, was adjusted on the pressure gauge. Afterwards, the lateral loading of the samples was performed through the control box, as shown in figure 3*a*. In the lateral loading process, the lateral pressure and displacement were monitored and recorded in real time.

## 3.4. Axial loading

After lateral loading, the upper cover plate was removed and the height of the samples recovered to slightly higher than the loading height (200 mm) of the samples; therefore, load was applied to the samples to pre-compress the samples to a height of 200 mm again. Then, the pressure for axial loading was set, followed by axial loading as shown in figure 3*b*. The lateral pressure, axial pressure and axial displacement were monitored and recorded in real time.

## 3.5. Grading after loading

The samples were graded through square-aperture sieves after axial loading. The particle size distributions of the samples before and after loading were recorded (figure 4).

# 4. Computation of compaction parameters

## 4.1. Lateral strain and porosity

### 4.1.1. Transformation from lateral stress to oil pressure in lateral loading

The lateral loading on the samples of broken waste rocks is realized by virtue of oil pressure provided by the oil cylinder for loading. The lateral stress in the test scheme refers to the compressive stress borne by the samples. Therefore, it is necessary to equate the lateral stress to an oil pressure. Then, lateral loading is applied to samples of broken waste rocks under different lateral stresses by setting the oil pressure on the pressure gauge. The relationship between oil pressure ($\sigma_o$) and lateral stress ($\sigma_h$) is

$$\sigma_o = \frac{\sigma_h A_h}{A_o} = \frac{\sigma_h L_h h_h}{\pi r_o^2}, \tag{4.1}$$

where $A_h$, $L_h$ and $h_h$ represent the area, length and height of the side-push plate, respectively; $A_o$ and $r_o$ refer to the cross-sectional area and the radius of the oil cylinder used for loading.

As the length and height of the side-push plate, and radius of the oil cylinder used for loading are known, they can be substituted in formula (4.1) to find that the oil pressure should be 6.5 MPa corresponding to a lateral stress of 2 MPa.

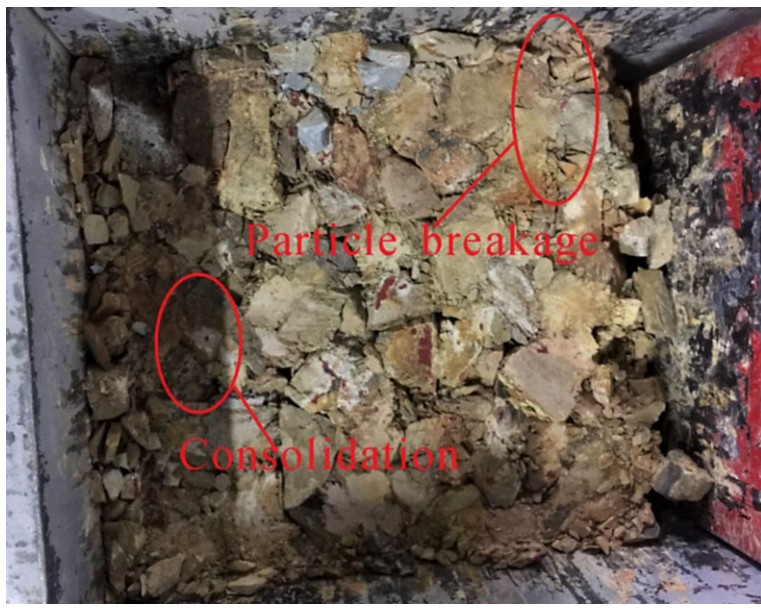

**Figure 4.** Samples after compaction.

### 4.1.2. Computing the lateral strain on samples during lateral loading

The lateral stress ($\sigma_h$) on samples during lateral loading is the ratio of the lateral loading pressure to the area of the side-push plate, and is calculated thus

$$\sigma_h = \frac{P_h}{A_h} = \frac{\sigma_o A_o}{A_h} = \frac{\sigma_o \pi r_o^2}{L_h h_h}, \tag{4.2}$$

where $P_h$ denotes the lateral pressure on the samples.

The lateral strain ($\varepsilon_h$) of samples during lateral loading can be expressed as the ratio of displacement resulting from lateral loading to the length of the loaded region

$$\varepsilon_h = \frac{\Delta L_h}{L_s}, \tag{4.3}$$

where $\Delta L_h$ and $L_s$ denote the displacement of samples due to lateral loading and the length of the loaded region, respectively.

### 4.1.3. Computing the lateral porosity of samples during lateral loading

Under lateral load, the porosity will change. The lateral porosity of samples during lateral loading is expressed as

$$\phi_h = \frac{V_h - V_0}{V_h} = 1 - \frac{m_s}{\rho_s(L_s - \Delta L_h)l_s h_s}, \tag{4.4}$$

where $V_h$ and $V_0$ refer to the volumes of the samples during lateral loading, and before being broken, respectively. In addition, $m_s$ refers to the mass of the samples and $\rho_s$ denotes the mass density of the samples. Moreover, $l_s$ and $h_s$ represent the width of the loaded region and the load height, separately.

## 4.2. Axial strain and porosity

### 4.2.1. Calculating the axial strain in samples during axial loading

The axial stress ($\sigma_v$) on samples during axial loading is the ratio of the axial load to the area of the upper compression plate. The area of the upper compression plate can be determined depending on lateral displacement. The axial stress can be calculated as follows:

$$\sigma_v = \frac{P_v}{A_v} = \frac{P_v}{L_v l_v} = \frac{P_v}{(L_s - \Delta L_h)l_v}, \tag{4.5}$$

where $P_v$ refers to the axial pressure applied to the samples. $A_v$, $L_v$ and $l_v$ are the area, length and width of the upper compression plate, respectively. In addition, $L_v$ is determined by the displacement of samples due to lateral loading.

The axial strain ($\varepsilon_v$) in samples under axial load is the ratio of displacement resulting from axial loading to the loading height, expressed as follows:

$$\varepsilon_v = \frac{\Delta h_v}{h_s}, \tag{4.6}$$

where $\Delta h_v$ represents the displacement of the samples under axial load.

### 4.2.2. Calculating axial porosity of samples during axial loading

The porosity of samples of broken waste rocks changes during axial loading. The axial porosity $\phi_v$ of the samples during axial loading is given by

$$\phi_v = \frac{V_v - V_0}{V_v} = 1 - \frac{m_s}{\rho_s(L_s - \Delta L_h)l_s(h_s - \Delta h_v)}, \tag{4.7}$$

where $V_v$ represents the volume of the samples under axial load.

# 5. Test results and discussion

## 5.1. Influences of lithology on lateral strain and porosity

According to the experimental data pertaining to the lateral loading on the samples and the results of using formulae (4.2)–(4.4), the changes in strain and porosity of the samples of different types of broken waste rocks under lateral loading are obtained (figures 5 and 6).

Analysis of the data in figures 5 and 6 reveals that:

(1) After the first lateral loading cycle, the lateral strain in samples of broken waste rock increased the most, then, as the number of cycles of lateral loading increased, the increase in lateral strain decreased as did the lateral porosity. This indicates that lateral loading can reduce the porosity and densify such waste rock samples. While, with increasing cyclic lateral compaction, the influences of lateral loading on lateral strain and lateral porosity decreased.

(2) The samples of the four types of broken waste rocks all showed elastic rebound after each cycle of lateral loading and unloading; however, the recovery only accounted for a low proportion of the total lateral deformation, while the lateral plastic deformation (irrecoverable deformation) was significant.

(3) The samples of lower strength were found to have larger decrease in lateral porosity. For example, the lateral porosity of broken shale fell from 0.425 to 0.344. In comparison, the decreases in lateral porosity of broken sandstone, mudstone and limestone samples were 0.044, 0.063 and 0.057, respectively, which indicated that broken sandstone underwent less lateral deformation, while broken shale underwent more lateral deformation, during lateral loading.

(4) The samples of higher strength underwent less lateral strain. The four types of broken waste rocks were listed, in descending order of lateral strain, as follows: shale, mudstone, limestone and sandstone. For instance, the lateral strain in broken sandstone samples was only 0.068, while that in broken shale samples was 0.122. This is because, due to the high strength of broken sandstone samples, the particles are stronger, stiffer and better able to resist slippage and rotation during lateral load; while particles in the broken shale samples were apt to be broken and slip, resulting in larger lateral deformation.

## 5.2. Influences of lithology on the axial strain and porosity

In accordance with the experimental results of the axial loading on the samples and data calculation using formulae (4.5)–(4.7), the changes in strain and porosity of different types of broken waste rocks under axial loading are as shown in figures 7 and 8.

From figures 7 and 8, it can be concluded that:

(1) The axial porosity of the stronger samples decreased least. For example, the axial porosity of broken sandstone samples fell from 0.422 to 0.205, while the decreases in the axial porosity of broken

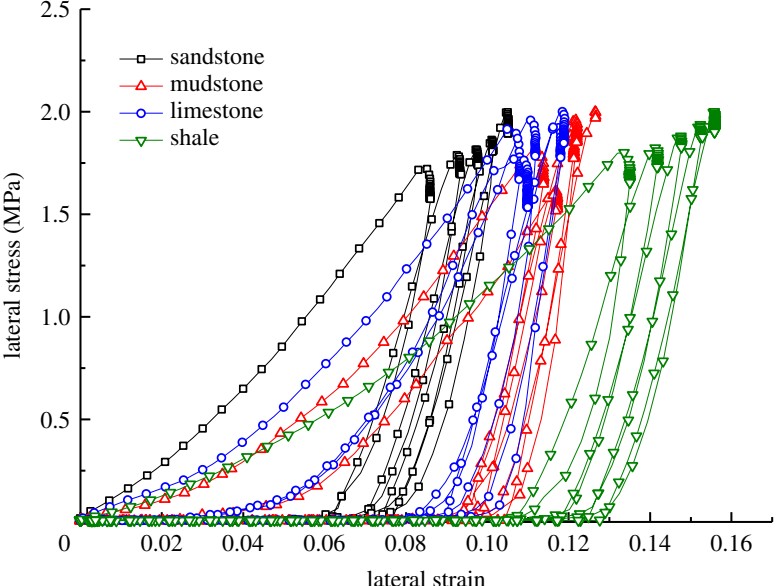

**Figure 5.** Changes in lateral strain of samples of different rock types.

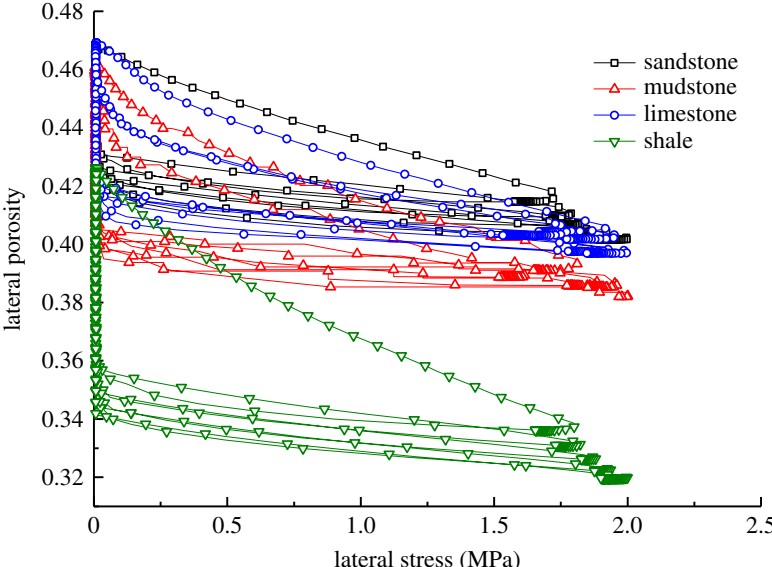

**Figure 6.** Changes in lateral porosity of samples of different rock types.

mudstone, limestone and shale samples reached 0.229, 0.225 and 0.237, respectively. The phenomenon indicates that the broken sandstone samples underwent the less axial deformation, while broken shale samples underwent the most axial deformation.

(2) The weaker samples underwent more axial strain. The four types of broken waste rocks were listed in descending order as shale, mudstone, limestone and sandstone in terms of the axial strain. To be specific, the axial strain in broken sandstone samples was 0.239, while that of broken shale samples was 0.295. This is because the higher strength of the broken sandstone samples endowed the particles thereof with a higher capacity to resist crushing, slippage and rotation during axial loading, while the particles of broken shale samples were easily broken and slipped, thus undergoing more axial deformation.

## 5.3. Influences of lithology on lateral stress and lateral pressure coefficient

Based on the experimental results of the samples during axial loading and the data arising from the use of formula (4.2), the changes in lateral stress and lateral pressure coefficient of the different types of

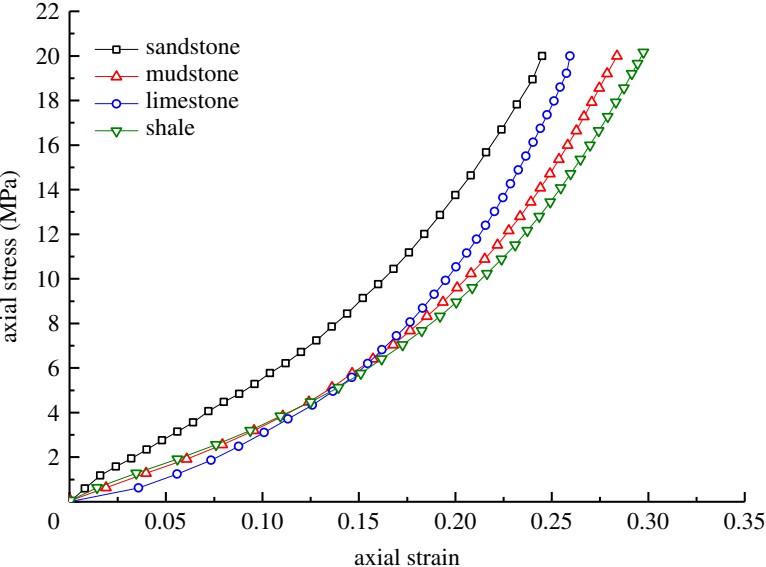

**Figure 7.** Changes in axial strain of samples of different rock types.

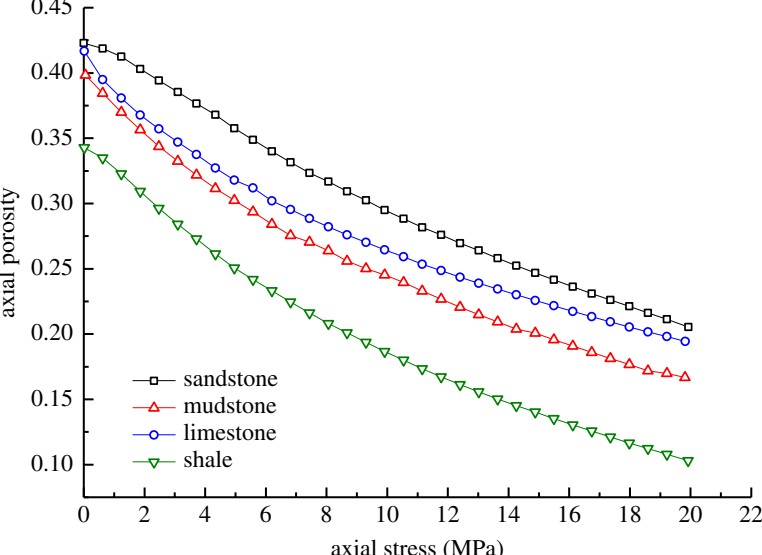

**Figure 8.** Changes in axial porosity of samples of different rock types.

broken waste rocks under axial loading are obtained (figures 9 and 10). The lateral pressure coefficient of the samples ($\eta$) is the ratio of the horizontal stress to vertical stress, as shown in the formula

$$\eta = \frac{\sigma_{mh}}{\sigma_{mv}}, \tag{5.1}$$

where $\sigma_{mh}$ and $\sigma_{mv}$ represent the horizontal and axial loading stress of the real-time monitoring, respectively.

Figures 9 and 10 show that:

(1) As the axial stress increased gradually, the lateral stress also increased, which can be roughly divided into three stages: rapid increase, increase at a decreasing rate and steady increase. This is because, as the axial stress gradually increased, the stress transferred to the lateral direction also increased.

(2) The lateral pressure coefficient first increased and then decreased under increasing axial stress. This is because, under low axial stresses unable to crush particles of the waste rocks used for backfill, most axial stress was transferred to the lateral direction; while under high axial stress, particles were

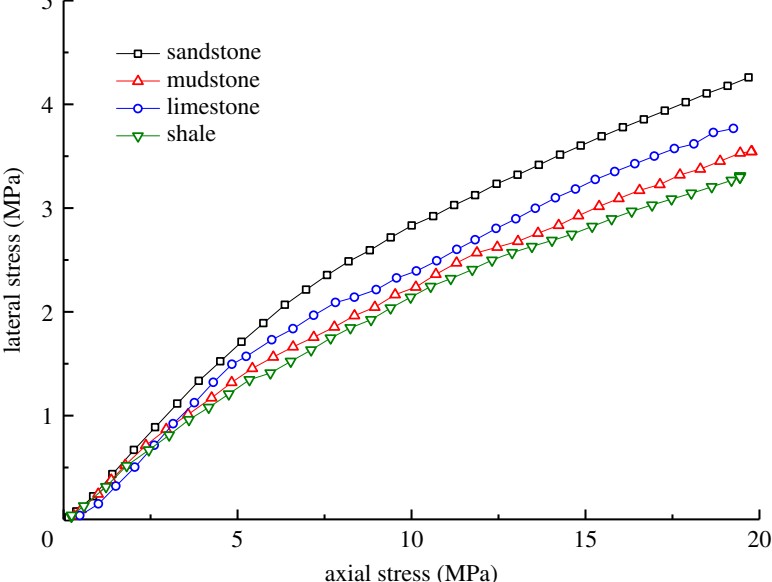

**Figure 9.** Changes in lateral stress of samples of different rock types.

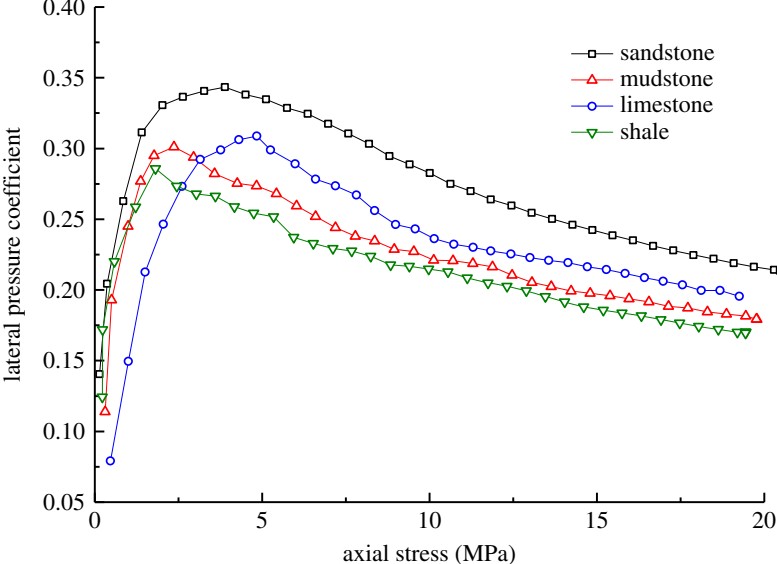

**Figure 10.** Changes in lateral pressure coefficient of samples of different rock types.

broken, thus dissipating energy from the work done by the testing machine. Finally, the lateral pressure coefficient declined slightly.

(3) The weaker samples presented a lower lateral stress. The four types of broken waste rocks are ranked, in descending order, as sandstone, limestone, mudstone and shale according to their lateral stress. To be specific, the lateral stress on the broken sandstone sample was as high as 4.24 MPa while that on the broken shale samples was only 3.31 MPa. This can be explained as follows: the broken sandstone samples were stronger, so their particles showed higher resistance to crushing, slippage and rotation during axial loading; in comparison, broken shale particles were apt to be broken and slip. As a result, compared with the broken shale samples, a higher stress was transferred to the lateral direction in broken sandstone samples.

(4) The stronger samples had a larger lateral pressure coefficient. For example, the lateral pressure coefficient of broken sandstone samples was 0.214, while those of broken mudstone, limestone and shale samples were 0.182, 0.195 and 0.168, respectively. The result suggested that a larger stress was applied to the lateral plane of the broken sandstone samples, and less to that in shale samples.

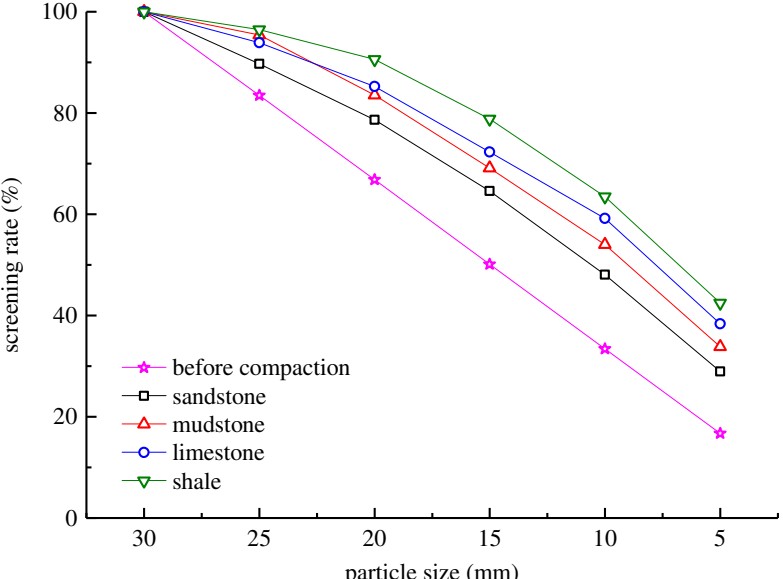

**Figure 11.** Particle size distributions before, and after, compaction.

## 5.4. Changes of particle size before, and after, compaction

The samples undergoing the compaction test were graded and the particle size distributions of the samples before, and after, loading were computed (figure 11).

As shown in figure 11, the particle size distributions of the four types of broken waste rocks all moved upwards relative to those before compaction. It indicates that the particles were broken and proportion of small particles increased constantly. Moreover, the higher the strength of a sample, the less the crushing: for instance, particles of sandstone samples were broken less, while those of shale samples were broken more. Crushing resulted in large compaction-induced deformation of the waste rock samples.

# 6. Conclusion

We investigated the influence of lithology on the compaction behaviours of broken waste rocks by using a WAW-1000D servo-controlled testing machine and a self-made bidirectional loading test system for granular materials. The following conclusions were obtained:

(1) The porosity of weaker samples fell during lateral and axial loading, indicating that the samples of stronger waste rocks showed more apparent deformation in lateral and axial loading. The four types of broken waste rocks were listed in a descending order as shale, mudstone, limestone and sandstone in terms of the decrease in porosity.

(2) The stronger samples underwent smaller lateral and axial strains. Samples of the four types of broken waste rocks all exhibited a spring-back effect during cyclic loading under lateral stress. Therein, the elastic deformation was small, while plastic deformations were large.

(3) During axial loading, the lateral stress on the four types of waste rocks changed in three stages: rapid increase, increase at a decreasing rate and steady increase, and the lateral pressure coefficient first increased and then decreased. Among the four types of waste rocks, stronger samples showed a higher lateral stress and lateral pressure coefficient.

(4) After compaction testing, the proportions of small particles in the four types of waste rocks all increased. In addition, for stronger samples, their particles were less severely broken. The four types of waste rocks, in descending order of crushing were: shale, mudstone, limestone and sandstone.

(5) By this paper's research, we can obtain the influence of lithology on the compaction behaviours of broken waste rock. In order to get the better backfill effects, the reasonable lithology can be optimized during backfill. The reasonable particle size can improve the support ability of backfill materials for strata and surface. Meanwhile, the waste rock dumps can be reduced, so the coal mine environment can be protected greatly.

Data accessibility. We include all the experimental data in the electronic supplementary material, which are available from the Dryad Digital Repository at: https://doi.org/10.5061/dryad.sj52ds3 [28].

Authors' contributions. M.L. prepared and edited the manuscript. J.Z. provided theoretical and methodological guidance in the research process. Z.W. and A.L. partially participated in the literature search and data processing. Y.L. participated in revising the manuscript.

Competing interests. We declare we have no competing interests.

Funding. This research was funded by the National Postdoctoral Program for Innovative Talents (BX20180361), the China Postdoctoral Science Foundation (2018M642366), the National Science Fund for Distinguished Young Scholars (51725403) and the State Key Laboratory of Coal Resources and Safe Mining, CUMT (SKLCRSM18KF009).

Acknowledgements. The authors are grateful for the valuable comments from the editors and the reviewer, which have substantially improved the quality of our work.

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
