## [Reviewer comments · Royal Society Open Science]

Review History

RSOS-182205.R0 (Original submission)

Review form: Reviewer 1

Is the manuscript scientifically sound in its present form?

Yes

Are the interpretations and conclusions justified by the results?

Yes

Is the language acceptable?

No

Is it clear how to access all supporting data?

Not Applicable

Do you have any ethical concerns with this paper?

No

Have you any concerns about statistical analyses in this paper?

No

Recommendation?

Accept with minor revision (please list in comments)

Comments to the Author(s)

The manuscript is easy to follow and has an appropriate structure. It is an interesting experimental study and might have applications in the coal mine environmental protection. However, there are some contents needing to be modified to improve the quality of the manuscript. Therefore, I suggest a minor revision before the manuscript could be accepted for publication. The following suggestions and comments should be considered when revising the manuscript.

Specific Comments:

No.1: the compaction of crushed waste rock was the main focus of this study, which is the main method to quantify the compaction process.

No.2: The lateral pressure applied was 2 MPa. Why the authors set the values for the lateral pressure and loading cycle?

No.3: Page 6 line 15, add an introductory sentence, before start describing the steps.

No.4: Figure 2 is similar to Figure 3. The authors should consider to merge into one figure.

No.5: In section 4, comparing Equations (1), (3), (4) and (5), it results that $h=hh$ and $h=$: why do the authors use different symbols for the same dimensions?

No. 6: Page 10 line 33, the authors should provide a formula defining the lateral pressure coefficient, please clarify.

No. 7: In the introduction and in the conclusions, greater emphasis must be placed on the fields of application of this research, not only about the experiment.

Review form: Reviewer 2

Is the manuscript scientifically sound in its present form?

Yes

Are the interpretations and conclusions justified by the results?

Yes

Is the language acceptable?

Yes

Is it clear how to access all supporting data?

Yes

Do you have any ethical concerns with this paper?

No

Have you any concerns about statistical analyses in this paper?

No

Recommendation?

Accept with minor revision (please list in comments)

Comments to the Author(s)

In this paper, a self-made bidirectional test system is used to test effects of lithology on compaction behaviors of waste rocks, which will provide the lithology selection for the backfill materials. Meanwhile, the research results will be helpful to reduce the pollutions in mine areas. The paper is interesting and is worthy of consideration for publication in the RSOS. However, many details are missing. The reviewer has some minor corrections or comments that the authors need to address to improve the quality of the paper.

According to the contents of sample preparation and test scheme, the waste rock sample is not belonging to coal gangue, which kind of the waste rock is used as the samples?

If this size of the loading box is enough for waste rock samples, please add the explanation.

Figure 4 is not necessary. The test steps have been stated in the main text. It should be deleted.

Page 6: line 23-24, after placing each layer, the rock blocks were pre-compacted until all layers were loaded. Not very clear. Please, rephrase or explain a bit better.

Page 10: line 32-34, by the term lateral pressure coefficient you mean the ratio of horizontal to vertical stresses?

Porosity usually represents the ratio between the volume of void and the total volume of a given material, containing also this void. On the other hand, h and ϵ in Equations (4) and (7) represent something similar to the volumetric strain $(-)/$, which is the ratio between the variation of volume and the original volume.

Decision letter (RSOS-182205.R0)

18-Feb-2019

Dear Professor Zhang,

The editors assigned to your paper ("An experimental study of the influences of lithology on compaction behaviours of broken waste rocks in coal mine backfilling") have now received comments from reviewers. We would like you to revise your paper in accordance with the referee and Associate Editor suggestions which can be found below (not including confidential reports to the Editor). Please note this decision does not guarantee eventual acceptance.

Please submit a copy of your revised paper before 13-Mar-2019. Please note that the revision deadline will expire at 00.00am on this date. If we do not hear from you within this time then it will be assumed that the paper has been withdrawn. In exceptional circumstances, extensions may be possible if agreed with the Editorial Office in advance. We do not allow multiple rounds of revision so we urge you to make every effort to fully address all of the comments at this stage. If deemed necessary by the Editors, your manuscript will be sent back to one or more of the original reviewers for assessment. If the original reviewers are not available, we may invite new reviewers.

When submitting your revised manuscript, you must respond to the comments made by the referees and upload a file "Response to Referees" in "Section 6 - File Upload". Please use this to

document how you have responded to the comments, and the adjustments you have made. In order to expedite the processing of the revised manuscript, please be as specific as possible in your response.

- Data accessibility

If you wish to submit your supporting data or code to Dryad (<http://datadryad.org/>), or modify your current submission to dryad, please use the following link:
<http://datadryad.org/submit?journalID=RSOS&manu=RSOS-182205>

- Competing interests

- Authors' contributions

- Acknowledgements

- Funding statement

on behalf of Prof R. Kerry Rowe (Subject Editor)
 openscience@royalsociety.org

Associate Editor's comments:

Please revise your manuscript to fully address the reviewers' concerns. They are broadly positive about your work, but we would like you to make sure you do a good job with the revisions (both scientifically and in checking the language in case you can improve its clarity at all).

The reviewers may be asked to assess your changes, so make sure you provide a full point-by-point response in your revision. Good luck!

Comments to Author:

Reviewers' Comments to Author:

Reviewer: 1

Comments to the Author(s)

The manuscript is easy to follow and has an appropriate structure. It is an interesting experimental study and might have applications in the coal mine environmental protection. However, there are some contents needing to be modified to improve the quality of the manuscript. Therefore, I suggest a minor revision before the manuscript could be accepted for publication. The following suggestions and comments should be considered when revising the manuscript.

Specific Comments:

No.1: the compaction of crushed waste rock was the main focus of this study, which is the main method to quantify the compaction process.

No.2: The lateral pressure applied was 2 MPa. Why the authors set the values for the lateral pressure and loading cycle?

No.3: Page 6 line 15, add an introductory sentence, before start describing the steps.

No.4: Figure 2 is similar to Figure 3. The authors should consider to merge into one figure.

No.5: In section 4, comparing Equations (1), (3), (4) and (5), it results that $h=hh$ and $h==$: why do the authors use different symbols for the same dimensions?

No. 6: Page 10 line 33, the authors should provide a formula defining the lateral pressure coefficient, please clarify.

No. 7: In the introduction and in the conclusions, greater emphasis must be placed on the fields of application of this research, not only about the experiment.

Reviewer: 2

Comments to the Author(s)

In this paper, a self-made bidirectional test system is used to test effects of lithology on compaction behaviors of waste rocks, which will provide the lithology selection for the backfill materials. Meanwhile, the research results will be helpful to reduce the pollutions in mine areas. The paper is interesting and is worthy of consideration for publication in the RSOS. However, many details are missing. The reviewer has some minor corrections or comments that the authors need to address to improve the quality of the paper.

According to the contents of sample preparation and test scheme, the waste rock sample is not belonging to coal gangue, which kind of the waste rock is used as the samples?

If this size of the loading box is enough for waste rock samples, please add the explanation.

Figure 4 is not necessary. The test steps have been stated in the main text. It should be deleted.

Page 6: line 23-24, after placing each layer, the rock blocks were pre-compacted until all layers were loaded. Not very clear. Please, rephrase or explain a bit better.

Page 10: line 32-34, by the term lateral pressure coefficient you mean the ratio of horizontal to vertical stresses?

Porosity usually represents the ratio between the volume of void and the total volume of a given material, containing also this void. On the other hand, h and ϵ in Equations (4) and (7) represent something similar to the volumetric strain $(-)/$, which is the ratio between the variation of volume and the original volume.

Author's Response to Decision Letter for (RSOS-182205.R0)

See Appendix A.

RSOS-182205.R1 (Revision)

Review form: Reviewer 1

Is the manuscript scientifically sound in its present form?

Yes

Are the interpretations and conclusions justified by the results?

Yes

Is the language acceptable?

Yes

Is it clear how to access all supporting data?

Yes

Do you have any ethical concerns with this paper?

No

Have you any concerns about statistical analyses in this paper?

No

Recommendation?

Accept as is

Comments to the Author(s)

None

Review form: Reviewer 2

Is the manuscript scientifically sound in its present form?

Yes

Are the interpretations and conclusions justified by the results?

Yes

Is the language acceptable?

Yes

Is it clear how to access all supporting data?

Not Applicable

Do you have any ethical concerns with this paper?

No

Have you any concerns about statistical analyses in this paper?

No

Recommendation?

Accept as is

Comments to the Author(s)

The manuscript can be published in its present form. Congratulations to the authors.

Decision letter (RSOS-182205.R1)

11-Mar-2019

Dear Professor Zhang,

I am pleased to inform you that your manuscript entitled "An experimental study of the influence of lithology on compaction behaviour of broken waste rock in coal mine backfill" is now accepted for publication in Royal Society Open Science.

on behalf of Professor R. Kerry Rowe (Subject Editor)
openscience@royalsociety.org

Associate Editor Comments to Author:

As the reviewers are now satisfied with your manuscript, it may be accepted for publication. Well done on your hard work!

Reviewer comments to Author:

Reviewer: 2

Comments to the Author(s)

The manuscript can be published in its present form. Congratulations to the authors.

Reviewer: 1

Comments to the Author(s)

None

Appendix A

Dear editor and reviewers,

On behalf of my co-authors, we thank you very much for giving us an opportunity to revise our manuscript. We are grateful to the editor and reviewers for their positive and constructive comments and suggestions on our manuscript entitled “An experimental study of the influence of lithology on compaction behaviour of broken waste rock in coal mine backfill” (RSOS-182205). These comments have all been of great help to us in the revision and improvement of our paper, as well as providing significant guidance for our research.

We have revised our manuscript according to the comments from the reviewers and have used a professional language editing service to improve its grammar and writing style. We hope that the manuscript now meets with your approval. The main corrections to the paper and our responses to the reviewers are as follows:

Responds to Reviewer #1

No. 1: The compaction of crushed waste rock was the main focus of this study, which is the main method to quantify the compaction process.

Response: Thank you for your positive comments. The “Method of compaction testing of solid backfilling materials” is used as the testing method for the compaction characteristics of broken waste rocks, which was issued by the China’s National Energy Administration. The authors have added this standard as the reference in the revised manuscript.

No. 2: The lateral pressure applied was 2 MPa. Why the authors set the values for the lateral pressure and loading cycle?

Response: Thank you for your positive comments. The lateral pressure (2 MPa) was set according to the engineering applications.

No. 3: Page 6 line 15, add an introductory sentence, before start describing the steps.

Response: Thank you for your positive comments. The authors have added the introductory sentence.

No. 4: Figure 2 is similar to Figure 3. The authors should consider to merge into one figure.

Response: Thank you for your positive comments. The authors have merged Figure 2 and Figure 3 into one Figure in the revised manuscript.

No. 5: In section 4, comparing Equations (1), (3), (4) and (5), it results that $h_s=h_h$ and $L_h=l_s=l_v$: why do the authors use different symbols for the same dimensions?

Response: Thank you for your positive comments. The authors used different symbols for the different dimensions.

No. 6: Page 10 line 33, the authors should provide a formula defining the lateral pressure coefficient, please clarify.

Response: Thank you for your positive comments. The authors have added the formula in the revised manuscript.

No. 7: In the introduction and in the conclusions, greater emphasis must be placed on the fields of application of this research, not only about the experiment.

Response: Thank you for your positive comments. The authors tried their best to add more detailed information about the fields of application of this paper's research.

Special thanks to you for your good comments again. They are valuable and very helpful for revising and improving our paper, as well as the important guiding significance to our researches.

Responds to Reviewer #2

No. 1: According to the contents of sample preparation and test scheme, the waste rock sample is not belonging to coal gangue, which kind of the waste rock is used as the samples?

Response: Thank you for your positive comments. The waste rock produced during tunneling was used as the testing samples.

No. 2: If this size of the loading box is enough for waste rock samples, please add the explanation.

Response: Thank you for your positive comments. The maximum particle size of the waste rock sample is 30mm. And the minimum size of the loading box is 200mm. The ratio of the minimum size of the loading box to the maximum particle size of the waste rock sample is bigger than 6.

No. 3: Figure 4 is not necessary. The test steps have been stated in the main text. It should be deleted.

Response: Thank you for your positive comments. The authors have deleted Figure 4 in the revised manuscript.

No. 4: Page 6: line 23-24, after placing each layer, the rock blocks were pre-compacted until all layers were loaded. Not very clear. Please, rephrase or explain a bit better.

Response: Thank you for your positive comments. The authors have added more information in the revised manuscript about this sentence.

No. 5: Page 10: line 32-34, by the term lateral pressure coefficient you mean the ratio of horizontal to vertical stresses?

Response: Thank you for your positive comments. The lateral pressure coefficient is the ratio of horizontal to vertical stresses. The authors have added one formula in the revised manuscript.

No. 6: Porosity usually represents the ratio between the volume of void and the total

volume of a given material, containing also this void. On the other hand, ϕ_h and ϕ_v in Equations (4) and (7) represent something similar to the volumetric strain $(V_{final}-V_{initial})/V_{initial}$, which is the ratio between the variation of volume and the original volume.

Response: Thank you for your positive comments. The ϕ_h and ϕ_v are calculated by Formula (4) and (7), V_h or V_v represents the volume of specimen (broken), the V_0 represents the volume of specimen (not broken), $V_h - V_0$ or $V_v - V_0$ represents the volume of void.

Special thanks to you for your good comments again. They are valuable and very helpful for revising and improving our paper, as well as the important guiding significance to our researches.